**R E S C I E N C E  C**

Editorial / ML Reproducibility Challenge 2023

# [Re] Towards Understanding Biased Client Selection in Federated Learning

Anonymous

Edited by
(Editor)

Reviewed by
(Reviewer 1)
(Reviewer 2)

Received
—

Published
—

DOI
—

**Abstract**  Federated learning is a distributed optimization algorithm that enables cooperatively training a machine learning model on resource limited client nodes. Such a decentralized model training approach does not require data exchange from client devices to global servers, therefore protecting data privacy and enhancing the model's generalizability by training on heterogeneous data. In this work, we perform a reproducibility study of a recent paper "Towards understanding biased client selection in federated learning" [1]. We reproduce the majority of the various experiments to validate the claim of the original paper that a biased client selection strategy can significantly speed up the training convergence of federated learning compared to a conventional random client selection strategy. In addition to reproduction, we explored the performance of proposed algorithm with different hyperparameters. The implemented code, training metrics, hyperparameters and resulted models are open-sourced on DagsHub for easy reproduction.

**Keywords**  Federated Learning, Reproducibility Study, Image Classification, Sentiment Analysis, Logistic Regression, Multi-layer Perceptron.

## 1  Reproducibility Summary

### 1.1  Scope of Reproducibility

This study investigates the reproducibility of the paper "Towards understanding biased client selection in federated learning" by Yae Jee Cho, Jianyu Wang, and Gauri Joshi. The authors propose the Power-of-Choice algorithm, claiming it converges three times faster with a 10% higher test accuracy compared to the baseline random selection algorithm. Their experiments cover (a) quadratic model optimization, (b) logistic regression on a synthetic dataset, (c) multi-layer perceptron on FMNIST for image classification, and (d) multi-layer perceptron on the Senti140 dataset for text sentiment analysis.

### 1.2  Methodology

The paper offers pseudo-code for the Power-of-Choice algorithm, but the supplementary material lacks completeness and necessitates a multi-server setup. We address this by re-implementing the code for most experiments, optimizing computation on a single node for server-client communication. Using a laptop CPU and Purdue's cluster, we successfully reproduce key figures and explore performance improvements under varied hyperparameters, particularly focusing on different learning rates and local iteration numbers.

### 1.3  Results

Our federated learning experiments confirm the Power-of-Choice algorithm's faster convergence than the random selection algorithm across tasks like quadratic optimization

and logistic regression. These advantages persist with small learning rates, limited local iterations, and constrained communication rounds. Yet, the biased client selection algorithm falters without these conditions, and in extreme scenarios, like large learning rates and infinite rounds, the random selection algorithm outperforms Power-of-Choice. Our study outlines the pros and cons of biased client selection compared to the conventional random approach.

## 1.4 What was easy

The original paper provides a detailed description of the Power-of-Choice algorithm, clearly outlining dataset information and training procedures. The supplementary material includes code that serves as a solid foundation for reproducibility studies.

## 1.5 What was difficult

Implementing the Power-of-Choice algorithm and its variants posed various challenges, including the need for code adaptation from a distributed (original codebase) to a single CPU/GPU environment (our implementation). Unspecified hyperparameters and inconsistencies complicated experimental setups, while the biased client selection algorithm exhibited less significant performance gains in specific experiments. Ambiguities in the training loss specification further hindered direct result comparisons with the original paper.

## 1.6 Communication with original authors

Efforts to contact the authors via email have seen limited success. Issues include discrepancies in results, unresolved queries about negative loss values in Figures 4 and 5 of the original paper, and a few minor clarifications pertaining to their implementation.

## 2 Introduction

Federated learning is a distributed optimization algorithm that enables cooperatively training a machine learning model on resource-limited client nodes [2, 3, 4]. Such a decentralized model training approach does not require data exchange from client devices to global servers [5], therefore protects data privacy [6] and enhances the generalizability of the model by training on heterogeneous data. In the conventional federated learning scenario, the model is locally trained for a few iterations on selected clients and the aggregating server periodically updates the global model based on local model updates. The conventional federated learning algorithm relies on unbiased client selection for the server model update, which requires excessive communication between the server and clients [7]. In paper [1], the authors perform a comprehensive study of their proposed biased client selection algorithm, called Power-of-Choice (*pow-d*). The authors demonstrate that biased client selection towards higher local loss yields faster convergence and higher test accuracy than unbiased random client selection. In addition, the authors provide several variants of their algorithm to reduce the computation (*cpow-d*) and communication (*rpow-d*) costs. The adaptive selection skew variant (*adapow-d*) simultaneously increases the converging speed and minimizes the non-vanish bias, enhancing the robustness of the Power-of-Choice algorithm.

In this work, we study the reproducibility of the paper "Towards understanding biased client selection in federated learning" [1] by Yae Jee Cho, Jianyu Wang and Gauri Joshi. We aim to reproduce their federated learning experiments to verify the main claims of the paper and perform additional experiments to further assess their proposed algorithm Power-of-Choice. Our contributions are:

1. Developing modular implementation of the algorithm Power-of-Choice for client selection in federated learning.

2. Reproducing majority of the experiments included in the main text of the original paper given the resource constraints.

3. Performing ablation studies for Power-of-Choice algorithm under various hyper-paramters.

## 2.1 Scope of reproducibility

In the paper, the authors propose a client selection algorithm, called Power-of-Choice, for federated learning where the idea is to select clients with higher losses for each communication round. With extensive machine learning experiments, the authors claim that the Power-of-Choice algorithm converges faster and gives higher test accuracy than the baseline random selection algorithm proposed in [2]. The authors perform 5 different experiments to support their claim for fast convergence of Power-of-Choice. In each experiment where the high-level task is classification, the model and dataset are altered keeping the underlying algorithm consistent to show the efficacy of their algorithm under diverse domains. The experiments performed to support their claims include

1. Quadratic model optimization

2. Logistic regression on synthetic dataset

3. Multi-layer perceptron on FMNIST [8] for image classification

4. Convolution Neural Network on CIFAR10 for image classification

5. Multi-layer perceptron on Senti140 dataset[9] for text sentiment analysis

We perform all except (4) to verify their claim on faster convergence due to resource constraints.

The remainder of the paper is organized as follows: Section 3 serves as a pre-requisite for federated learning and explains the proposed Power-of-Choice method in detail. Section 4 presents our strategy for conducting a reproducible study. Section 5 highlights the experiments conducted in more detail. Finally, Section 6 concludes our main findings from the paper based on our assessment.

# 3 Client Selection in Federated Learning

Consider a federated learning setup with $K$ clients, where client $k$ has local dataset $B_k$ containing $D_k$ data samples. The central aggregating server can communicate with the clients and aims to minimize the collective loss $F(w)$ on the whole dataset.

$$F(w) = \frac{1}{\sum_{k=1}^{K} D_k} \sum_{k=1}^{k} \sum_{\xi \in B_k} f(w, \xi) = \sum_{k=1}^{K} p_k F_k(w) \tag{1}$$

Here, $f(w, \xi)$ is the loss function for sample $\xi$ and parameter w. $F_k = \frac{1}{|B_k|} \sum_{\xi \in B_k} f(w, \xi)$ and $P_k = \frac{D_k}{\sum_{k=1}^{K} D_k}$ are the local objective function and the fraction of data for client $k$, correspondingly.
The traditional algorithm to solve the above optimization task is federated averaging (FedAvg)[2]. The algorithm breaks the training task into communication rounds. For each communication round, the global server selects $m = CK$ clients for training, where C is the selection fraction between 0 and 1. The selected client set for $t$-th communication

round is $S^{(t)}$. Each selected client takes $\tau$ iterations of local statistical gradient descent (SGD)[10, 11, 12, 13] and updates the new local model to the server. Finally, the server updates the global model by aggregating the local model updates and broadcasting the global model to the clients. The global model update for each communication round is:

$$\bar{w}^{(t+1)} = \bar{w}^{(t)} - \eta_t \bar{g}^{(t)} = \bar{w}^{(t)} - \frac{\eta_t}{m} \sum_{k \in S^{(t)}} g_k(w_k^{(t)}, \xi_k^{(t)}) \tag{2}$$

where, $\bar{w}^{(t)}$ is the global model parameter and $\eta_t$ is the learning rate at the $t-$th communication round.

**Random Strategy** — In the above described algorithm, the strategy of client selection plays a key role in the training performance. The conventional federated learning algorithm applies an unbiased random selection strategy $\pi_{rand}$: m clients are randomly sampled from totally K clients with the probability of $p_k$ to construct $S^{(t)}$. $\pi_{rand}$ guarantees unbiased converges, however, shows slow converge speed for heterogeneously distributed client datasets, which is the case for practical implementation of federated learning[14, 15].

**Power-of-Choice Strategy** — To increase the convergence speed, the author used a biased client selection strategy.

$$S^{(t+1)} = argmax_{k \in S_d^{(t+1)}} F_k(\bar{w}^{(t)}) \tag{3}$$

Here, $S_d^{(t+1)}$ is the d candidate clients set sampled without replacement from a total of $K$ clients with probability $p_k$. $F_k(\bar{w}^{(t)})$ is the local client loss. $S^{(t+1)}$ is constructed by selecting $m$ clients with the largest client loss. The author claims that this Power-of-choice ($\pi_{pow-d}$) algorithm can speed up the training convergence by training the model on selected high loss yielding clients.

In addition to $\pi_{pow-d}$, the authors proposed three variants algorithm to accommodate practical considerations[16, 17, 18].

1. $\pi_{cpow-d}$, saving computation cost by estimating local loss on a mini-batch of the client dataset.

2. $\pi_{rpow-d}$, saving both computation and communication cost by updating accumulated average loss over local iterations.

3. $\pi_{adapow-d}$, minimizing the bias by gradually reducing candidate client set until $d = m$, which approaches the unbiased selection $\pi_{rand}$.

## 4 Methodology

The authors of the paper provide pseudo-code for the Power-of-Choice algorithm and its variants. The code included in the supplementary material is not complete and requires the use of a multi-server environment. Several important hyperparameters of the experiments are not specified. In this work, we re-implement the code for most of the experiments. We run our code on laptop CPU and cluster to validate the claims of the paper. We are able to reproduce figures 2, 3, 4, and 6 of the main text. In addition to reproduction, we explore performance improvement of the Power-of-choice algorithm for various hyperparameters. We found out that the improvement of convergence speed from Power-of-Choice is more significant when the learning rate is small.

## 4.1 Experiments

To validate the claims in the paper, the authors performed various federated learning tasks on various models and datasets, as summarized below:

**Experiment 1: Quadratic Model Optimization** – In this task, the objective is to optimize over a linear combination of quadratic functions $F(w)$, wherein each constituent quadratic function represents a loss function $F_k(w)$ at client $k$. The quadratic functions are designed synthetically as further described in next section. Note that since we are directly optimizing over mathematical functions, we don't need data.

**Experiment 2: Logistic Regression on Synthetic Data** – In this task, a plain logistic regression model is used for classification task in conjunction with a synthetically generated federated dataset `Synthetic(1,1)` [19]. The data consists of 100 users (clients) and 10 classes with data distributed in a non-iid manner. The loss function used is cross entropy.

**Experiment 3: MLP based Image Classification** – Similar to above task, this task aims to learn a multi-layer perception (MLP) model using FMNIST dataset [8] using a negative log-likelihood loss function. The data is heterogeneously split across users in a non-iid manner using Dirichlet distribution ($Dir(\alpha)$) where $\alpha$ controls the amount of heterogeneity in the data split.

**Experiment 4: MLP based Sentiment Analysis** – As the name suggests, in this task, an MLP model is trained on Twitter dataset [9] using a negative log-likelihood loss function. The data is inherently split among users and we randomly filter 314 users with more than 32 tweets for our experiment due to large userbase with imbalanced data.

All the hyperparameters and model specifications have been used as specified in the original paper.

## 4.2 Computational requirements

The training experiment for quadratic optimization and MLP sentiment analysis are conducted on a laptop with Intel i7-13700H CPU. The training experiment for MLP based image classification and logistic regression are conducted on Purdue cluster. Our reproducibility study comes at an estimated total computational cost of 100 CPU hours. As our experiment requires training multiple models for over 500 epochs, the required long CPU time make it impossible to record all training process with MLflow. For example, it takes 3 CPU hours to record one training process for quadratic optimization and an estimated 100 hours to record all the experiment for quadratic optimization. Most of the CPU time are used for communication instead of computation. As a result, we only record the most important federated learning experiment with MLflow.

# 5 Results

## 5.1 Quadratic Model Optimization

Quadratic model optimization is among the basic machine learning tasks and is an important test-bed for federated learning algorithms where the loss function is strongly convex and smooth for statistical gradient descent. Our implementation of quadratic model optimization tests the performance of the proposed federated learning algorithm

under ideal conditions. We show that candidate number d for $\pi_{pow-d}$ algorithm controls the trade-off between convergence speed and bias.

For quadratic model optimization, the local objective function is a convex function:

$$F_k(w) = \frac{1}{2}w^T H_k w - e_k^T w + \frac{1}{2}e_k^T H_k^{-1} e_k \tag{4}$$

Here, $H_k = h_k I$ is a diagonal matrix and $h_k$ is sampled from uniform distribution $U(1, 20)$. $e_k$ is a randomly generated vector, acting as the client data. The client data size $p_k$ follows power law distribution $P(x; a) = ax^{a-1}$, where $a$ determines the heterogeneity of the training data. The gradient descent update of local clients is given by

$$g_k(w_k^{(t)}) = H_k w_k^{(t)} - e_k \tag{5}$$

To reproduce the result shown in Figure 2 of the original paper, we applied the same hyperparameters as the paper: $\tau = 2, \nu = 5, \eta = 2 \times 10^{-5}$ and record the global loss as a function of communication round. For $\pi_{adapow-d}$, $d$ decreases by half from $d = K$ to $d = m$ for every 5000 communication rounds. The produced result is shown as Figure 1-4, which agrees well with the experiment result of the original paper.

As shown in Figure 1 and Figure 2, the convergence speed of $\pi_{pow-d}$ algorithm is faster than $\pi_{rand}$ when the communication rounds are limited (e.g. when the communication rounds are less than 5000). The convergence speed boosting is more significant for larger d, which indicates larger candidate clients set and more biased selection strategy. However, as the $\pi_{pow-d}$ algorithm includes biased sampling into the training process, the global loss at the final convergence is larger than $\pi_{rand}$, which indicates training with unbiased data. When the communication rounds are large enough (e.g. when the communication rounds are 15000), the global loss of $\pi_{pow-d}$ for larger d is larger than $\pi_{rand}$ due to the large bias. Figure 4 shows the bias and convergence speed trade-off for different algorithms and different hyperparameters. In particular, candidate set number d is a key parameter to tune the bias level of $\pi_{pow-d}$ algorithm.

To balance the convergence speed and final global loss of biased client selection algorithm, the author proposed adaptive client selection strategy ($\pi_{adapow-d}$). As we mentioned before, while larger d indicates larger bias and higher initial convergence speed, smaller d leads to smaller final global loss. As d approaches m, the difference $\pi_{pow-d}$ and $\pi_{rand}$ vanishes. $\pi_{adapow-d}$ gradually reduce d, taking advantage of the fast initial convergence speed of biased selection algorithm and low final global loss of the unbiased selection algorithm. As a result, $\pi_{adapow-d}$ achieve better performance than $\pi_{rand}$ for a wide selection of communication rounds.

To validate the performance of $\pi_{adapow-d}$, we conducted federated learning experiment for $\pi_{rand}$, $\pi_{pow-d}$ with a small d and $\pi_{adapow-d}$, shown in figure 3. At the limit of large communication rounds, global loss of $\pi_{rand}$ is less than $\pi_{pow-d}$ with a small d by 2.5 orders of magnitude. At the same time, $\pi_{adapow-d}$ holds a constant advantage compared to $\pi_{rand}$ for a large communication rounds regime from 5000 to 45000.

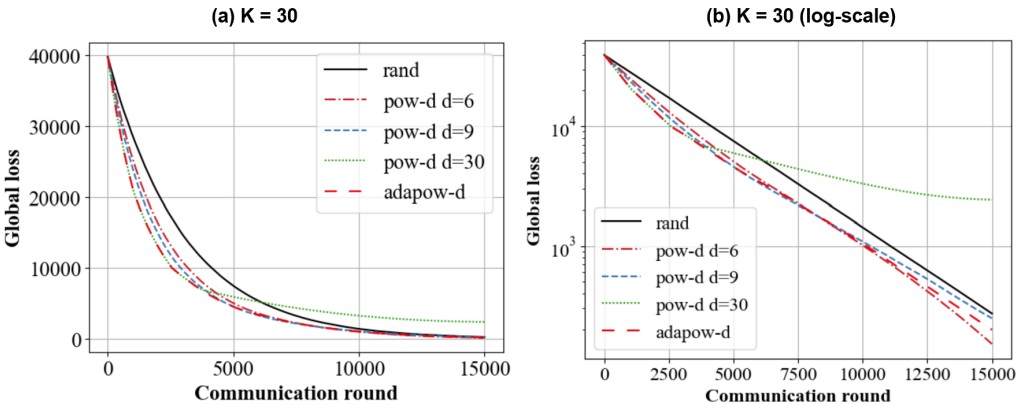

**Figure 1**. **Quadratic model optimization.** Global loss of 1500 communication rounds for three client selection strategies: $\pi_{rand}$, $\pi_{pow-d}$, $\pi_{adapow-d}$ with $C = 0.1$. (a) shows the global loss during the training process for the case of $K = 30$ clients. (b) shows the same loss curve in log scale.

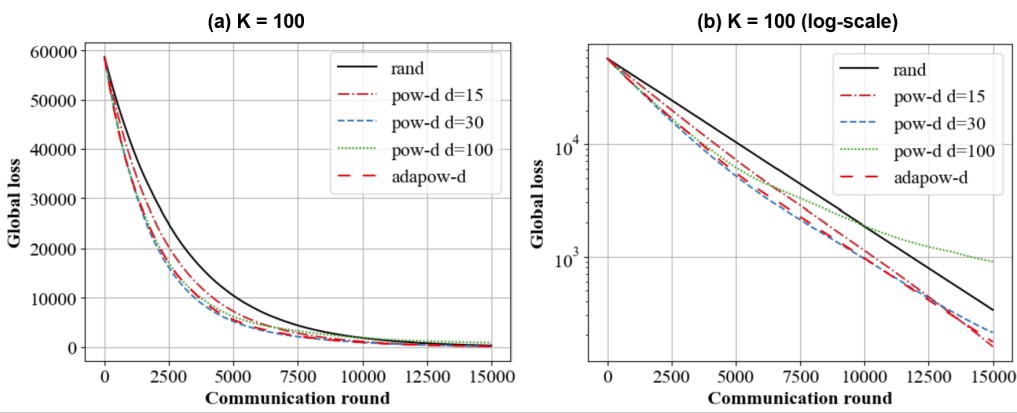

**Figure 2**. **Quadratic model optimization.** Global loss of 1500 communication rounds for three client selection strategies: $\pi_{rand}$, $\pi_{pow-d}$, $\pi_{adapow-d}$ with $C = 0.1$. (a) shows the global loss during the training process for the case of $K = 100$ clients. (b) shows the same loss curve in log scale.

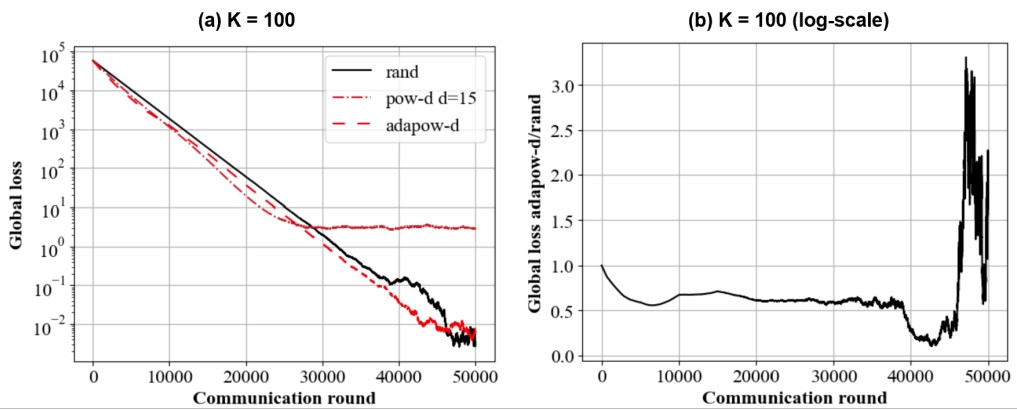

**Figure 3**. **Quadratic model optimization.** Global loss of 50000 communication rounds for three client selection strategies: $\pi_{rand}$, $\pi_{pow-d}$, $\pi_{adapow-d}$ with $C = 0.1$. (a) shows the global loss during the training process for the case of $K = 100$ clients. (b) shows the global loss ratio between $\pi_{adapow-d}$ and $\pi_{rand}$. The loss of $\pi_{adapow-d}$ is 40% smaller than $\pi_{rand}$ for moderate communication rounds from 5000 to 45000. However, after 45000 communication rounds, the results from the two algorithms are similar.

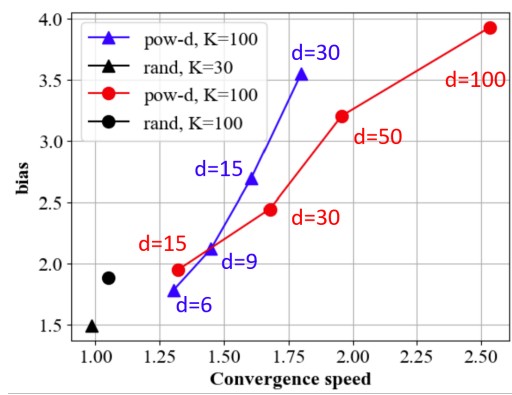

**Figure 4.** **Quadratic model optimization.** Bias vs Convergence speed for different algorithms and hyperparameters. The random client selection algorithm show slower convergence speed but a smaller bias than the base Power-of-Choice algorithm. For the same Power-of-Choice algorithm, larger candidate set number d shows faster convergence speed but larger bias in the end.

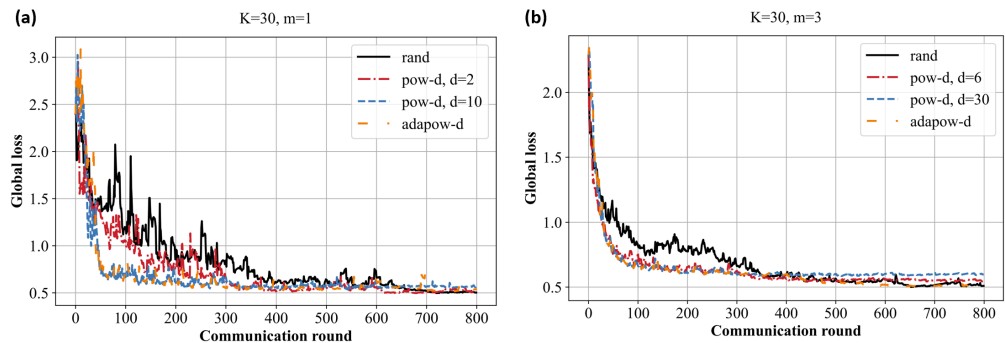

**Figure 5.** **Logistic regression on Synthetic(1, 1).** Three client selection algorithms $\pi_{rand}$, $\pi_{pow-d}$ and $\pi_{adapow-d}$ are compared against the same federated learning task with K=30. The result agrees well with Figure 3 of the original paper. There is no significant difference for the final test accuracy.

## 5.2 Logistic regression on Synthetic federated dataset

For this experiment, a relatively simple logistic regression model is trained on a synthetically generated federated dataset `Synthetic(1,1)` [19]. We use the same hyperparameters as mentioned in the original paper for this experiment and found a very good match with their results. The results are shown in Figure 5. These figures show the faster convergence of the proposed Power-of-Choice-d (pow-d) method and also suggest that a higher value of $d$ leads to a slightly sub-optimal solution as can be seen through the increased gap between the loss values. The corresponding comparison of accuracy values is shown in Figure 6.

## 5.3 MLP for image classification on FMNIST dataset

The authors also show the usefulness of their algorithm, not only on synthetic but real-world datasets. The dataset used is FashionMNIST and is split among 100 clients in a non-iid manner using Dirichlet Distribution ($Dir(\alpha)$) where the parameter $\alpha$ controls the degree of heterogeneity in the data. The task is that of image classification and the model used for training is MLP with two hidden layers. However, we find a departure

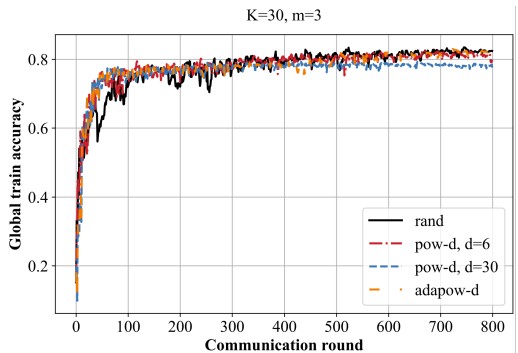

**Figure 6**. Logistic regression on Synthetic(1, 1). Test accuracy of three client selection algorithms $\pi_{rand}$, $\pi_{pow-d}$ and $\pi_{adapow-d}$ are compared against the same federated learning task with K=30.

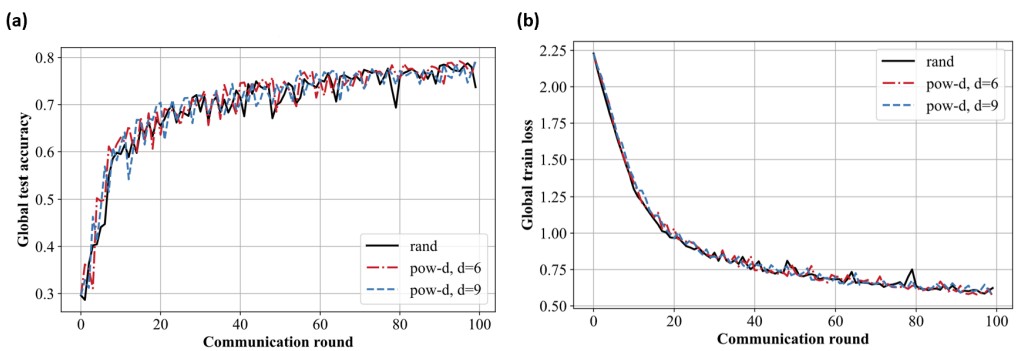

**Figure 7**. MLP for image classification on FMNIST dataset. For $\pi_{pow-d}$ with K = 100, C = 0.03 and varying d, learning rate $\eta = 0.005$. In our implementation, there is no significant difference in both convergence speed and final test accuracy for different algorithms.

from our results for those claimed in the paper. Figure 7 shows the closest result we could get for the same hyperparameters. We tried other values of learning rates but none of them produced desired results. Our implementation seemed to work with a synthetic dataset but when the dataset and model are switched to a real-world use case, the effectiveness of the proposed approach is not apparent. We find out that the random sampling approach is equally better and all of them beat the accuracy values claimed in the paper. We, however, see a slightly faster convergence than the random approach but the margin is very minimal. Moreover, we don't find a gap in the converged value of accuracy for various approaches but the paper seems to show an appreciable difference in them. We also test with reducing the learning rate to $\eta = 0.001$ but the results (Figure 8) don't seem to align with the original paper.

## 5.4 MLP for sentiment classification on Twitter dataset

Sentiment analysis predicts the emotional tone from texts, which is an widely used technique to help companies obtain feedback from clients and improve their products and services. Sentiment analysis algorithm can be applied for a variety of tasks such as brand monitoring, market research, and campaign performance tracking. Here our implementation shows that biased client selection strategy improves the convergence speed of sentiment analysis in federated learning tasks.

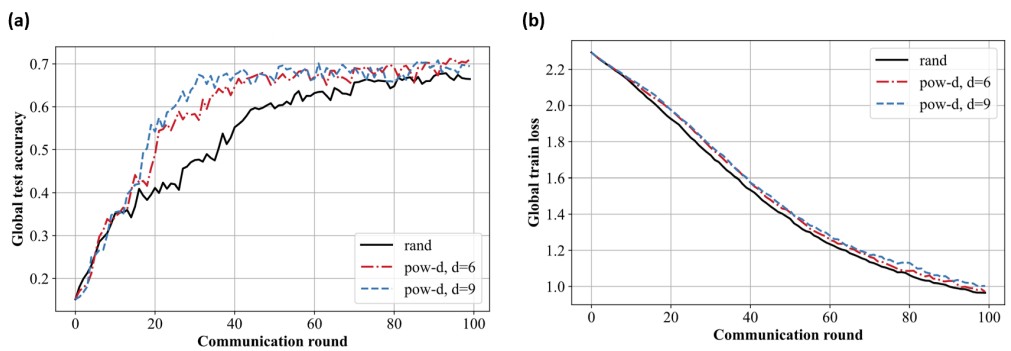

**Figure 8.** MLP for image classification on FMNIST dataset. For $\pi_{pow-d}$ with K = 100, C = 0.03 and varying d, learning rate $\eta = 0.001$. In our implementation, the convergence speed from $\pi_{pow-d}$ is higher than $\pi_{rand}$. There is no significant difference in the final reached test accuracy or training loss.

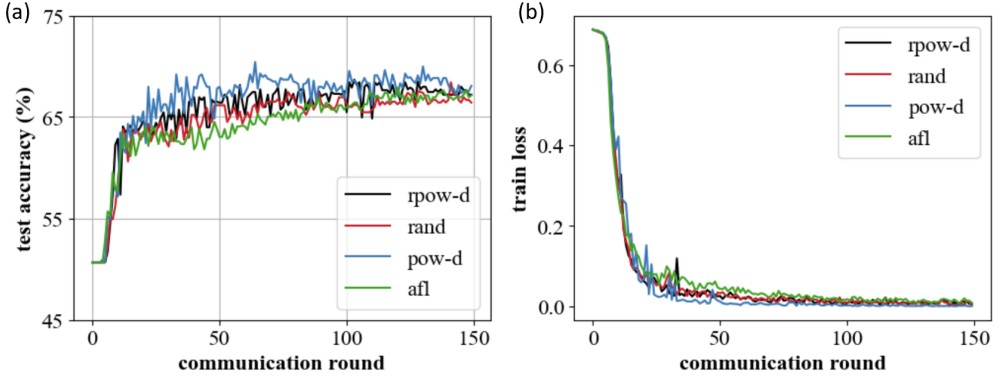

**Figure 9.** MLP for sentiment analysis Test accuracy as a function of communication round for $\pi_{pow-d}$, $\pi_{afl}$, $\pi_{rpow-d}$ and $\pi_{rand}$ with learning rate $\eta = 0.05$. d = 32 is chosen for the $\pi_{pow-d}$ algorithm.

We randomly select 314 users that have more than 32 tweets from the Twitter dataset. The data of each user are sent to individual clients, which naturally sets the data heterogeneity as each user has a different language habit/style. The heterogeneity across the clients can be further increased by reducing the number of tweets criteria when selecting the users. For each user, the data includes tweet content (text) and sentiment (positive and negative with label 0 and 1 respectively). Prior to the training, extensive text processing techniques are applied to clean the text for sentiment analysis. The data processing procedures including, tagging urls/users/emojis/numbers, removing punctuation, Stemming, etc. The cleaned tweet texts are embedded with a pretrained 200D Glove embedding[20] as the input of the model. The multi-layer perceptron (MLP) model for sentiment analysis has three hidden layers with units 128, 86 and 30.

In the original manuscript, the author claims to use $b = 32$ with $\tau = 100$ and $\eta = 0.05$. However, in our implementation, the difference between $\pi_{rand}$ and $\pi_{pow-d}$ is negligible using the same conditions, shown in figure 9. After conducting the experiments with different hyper-parameter combinations, we find that training with a smaller learning rate such as $\eta = 0.005$ shows more significant differences. From the test accuracy curve shown in figure 10, we observe increased convergence speed for Power-of-Choice client selection algorithm, which supports the claims of the paper.

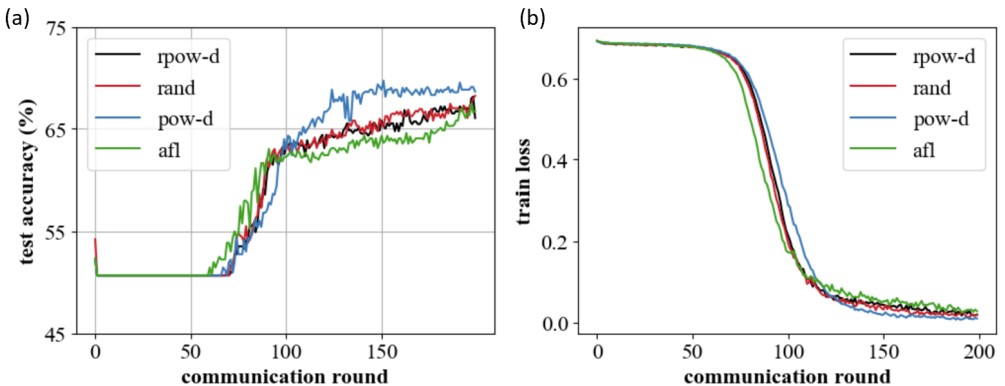

**Figure 10.** MLP for sentiment analysis Test accuracy as a function of communication round for $\pi_{pow-d}$, $\pi_{afl}$, $\pi_{rpow-d}$ and $\pi_{rand}$ with learning rate $\eta = 0.005$. d = 32 is chosen for the $\pi_{pow-d}$ algorithm.

## 6 Discussion

We performed comprehensive federated learning experiments to validate the claim of the paper [1] that biased client selection strategy can speed up the convergence compared to random client selection. Our implementation confirms that this advantage is valid under the condition of small learning rate, small local iterations, and limited communication rounds, which can be the case for a lot federated learning scenarios. We conducted further federated learning experiments and show that when the above three conditions are not met, the performance of Power-of-Choice degrades faster than the simple random client selection strategy. The conclusion is valid from simple quadratic optimization and logistic regression to complex MLP-based image classification and sentiment analysis tasks.

### 6.1 What was easy

The original paper describes the Power-of-Choice algorithm in detail. Information regarding the dataset and training procedures are clearly stated. The code provided in the supplementary material is a good starting point for the reproducibility study. Moreover, the machine learning model involved in the experiments is small and can be trained on a Laptop CPU with our implementation, enabling testing of the proposed algorithm with different hyperparameters.

### 6.2 What was difficult

(i) In addition to the Power-of-Choice algorithm, the author also provides several variation models, including computational-efficient variant, communication- and computation-efficient variant, adaptive selection skew variant. The four algorithms are compared against the random selection algorithm on five federated learning scenarios, leading to tons of replication experiments. (ii) The code in supplementary material requires a distributed system with multiple GPU/CPU, which is not available for most of researchers. We have re-implemented the code to make it work on a single CPU/GPU environment. (iii) Some of the hyperparameters for the model are not specified in the paper or are not consistent with the experiment result. (iv) The performance improvement of biased client selection algorithm on image classification and sentiment analysis experiments

is less significant compared to the original paper. (v) The training loss applied in the original paper is not clearly specified and is different from conventional negative log-likelihood loss (NLL) for classification tasks. This make it difficult to directly compared our results with that presented in the original paper.

### 6.3 Communication with original authors

We have tried reaching the authors through email but had very little communication. We posed the problem of differing results, primarily starting from Figure 4 onwards where all the methods (including the baseline) performed similar without any appreciable different in terms of convergence rates according to their specified hyperparameter configuration. Another major unresolved query was regarding the negative loss values in the Figures 4, 5, etc. The author responded that they used negative log-likelihood (NLL) for loss function. To our knowledge, NLL function produces always produces positive values given the input in terms of probability. We hope to hear back from the authors soon for a better resolution.

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

## 7 Appendix

Here Figure 11 and Figure 12 show examples of Fashion MNIST dataset and Twitter dataset for image classification and sentiment analysis tasks, respectively.

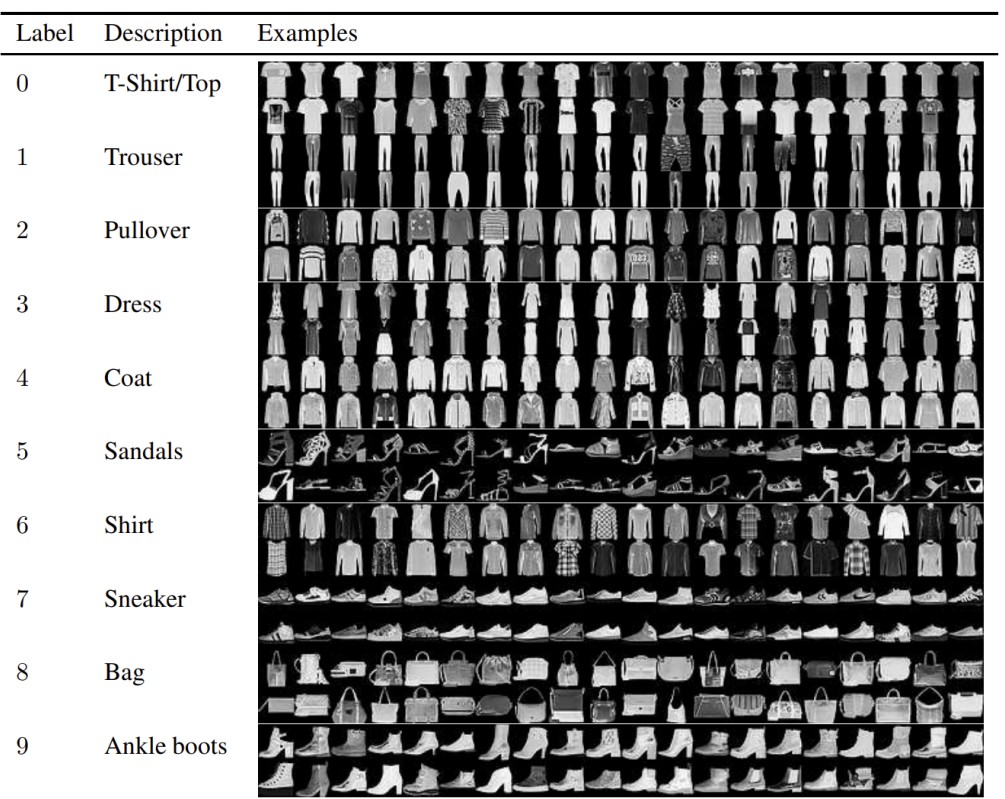

| Label | Description | Examples |
|-------|-------------|----------|
| 0 | T-Shirt/Top | |
| 1 | Trouser | |
| 2 | Pullover | |
| 3 | Dress | |
| 4 | Coat | |
| 5 | Sandals | |
| 6 | Shirt | |
| 7 | Sneaker | |
| 8 | Bag | |
| 9 | Ankle boots | |

**Figure 11. Fashion MNIST dataset [21].** The dataset consists of 60000 28×28 gray-scale images from 10 classes. Each class contains around 6000 images. The heterogeneous data partition among client is achieved with Dirichlet distribution $Dir_K(\alpha)$, where $\alpha$ controls data heterogeneity.

| user | tweet | Sentiment |
|---|---|---|
| peekinc | @mommygoggles Yay! Happy Thursday! | Positive |
| nathanhoad | My keyboard couldn't have picked a better time to die | Negative |
| mshawley | Totally relaxing today...reading, catching up on dvr'd shows, possibly a nap... | Positive |
| msbellee | @1capplegate And you make us smile as well as persevere. You are an inspiration to us all. | Positive |
| gemmacagnacci | had a dream all the snow got washed away by the rain | Negative |
| Rafaelcalvo | Watcing the dark knight its almost over.. | Negative |
| Linzzyy | caught a cold... huhuhuhu | Negative |
| homaygeefrawg | just got back from church. | Positive |
| EPMorgan | @HelenH20 ah I see! Well this is all cus I am so amaze! | Positive |
| shanajaca | @xXFriendXx HAHAHAHAHA you see it is possible i had that the other night disturbing right lol XX | Positive |

**Figure 12. Twitter dataset [9].** The dataset consists of 1600000 tweets with sentiment label. For the federated learning sentiment analysis experiment, we select 314 users with more than 32/64 tweets record. The tweets for each user are naturally heterogeneous data for the clients.

