# OpenReview forum: "[Re] Towards Understanding Biased Client Selection in Federated Learning"
_purdue.edu/Purdue_University/ML/2023/Hackathon_Reproducibility_Challenge — Purdue University ML 2023 Hackathon Reproducibility Challenge Submission_

### Official Review · Reviewer_5c7S · 2023-11-28
**Review of Toward Understanding Biased Client Selection in Federated Learning**

**Rating:** 8
**Confidence:** 4

**Review:**

**Quality**

Writing is occasionally colloquial "Figure 7 shows the closest result we could get for the same hyperparameters.", "leading to tons of replication experiments."

Section 5.3 "We tried other values of learning rates but none of them produced desired results." should have a graph for this claim.

Remove (a) and (b) from figures if they aren't referenced separately.

Authors claim in the discussion that "Our implementation confirms that this advantage is valid under the condition of small learning rate, small local iterations, and limited communication rounds." This would be a key insight, however, this should be visualized and given another subsection under section 5. Claims are unclear without the data.

Add an additional sentence or two justifying why single CPU/GPU experiments are comparable to the original author's multi CPU/GPU experiments.

**Clarity**

Reword the first couple of sentences in the abstract to make it clear that federated learning is decentralized.

Certain sections, such as 1.3-1.6, appear out of order. Unclear if this is due to author error or competition instructions.

Specify which cluster you used (Purdue University's XYZ Computing Cluster) before using the terms "Purdue cluster" or "cluster".

**Pros**

Overall great paper.

Great insights on the limitations of the algorithm in the results sections.

Replication seems to be faithful to the original paper, and the authors dove deep to explore the claims made by the original authors.

Great code quality with many comments, extremely easy to follow and read. All experiments were tracked using MLFlow.

**Cons**

Did not reproduce all experiments, but this is due to resource constraints. Authors reproduced 4/5 experiments.

Some overarching claims must be expanded upon further.

Writing style is occasionally too colloquial.